Comparative genomics reveals high genetic similarity among strains of Salmonella enterica serovar Infantis isolated from multiple sources in Brazil

Vilela Felipe P. 1
Felice Andrei G. 2
Seribelli Amanda A. 3
Rodrigues Dália P. 4
Soares Siomar C. 2
Allard Marc W. 5
Falcão Juliana P. jufalcao@fcfrp.usp.br 1
1 School of Pharmaceutical Sciences of Ribeirão Preto, Department of Clinical Analyses, Toxicology and Food Science, Universidade de São Paulo , Ribeirão Preto , São Paulo , Brazil
2 Institute of Biological and Natural Sciences, Department of Microbiology, Immunology and Parasitology, Universidade Federal do Triângulo Mineiro , Uberaba , Minas Gerais , Brazil
3 Medical School of Ribeirão Preto, Department of Cellular and Molecular Biology, Universidade de São Paulo , Ribeirão Preto , São Paulo , Brazil
4 Oswaldo Cruz Institute, Fundação Oswaldo Cruz , Rio de Janeiro , Rio de Janeiro , Brazil
5 Center for Food Safety and Applied Nutrition, U.S. Food and Drug Administration , College Park , MD , United States of America
Soares Paula
Electronic publication date: 2024 May 20
Publication date: 2024
Volume: 12
Electronic Location ID: e17306
Received 2023 Jun 2; Accepted 2024 Apr 4
Copyright: ©2024 Vilela et al.
Copyright year: 2024
Copyright holder: Vilela et al.
License: This is an open access article distributed under the terms of the Creative Commons Attribution License, which permits unrestricted use, distribution, reproduction and adaptation in any medium and for any purpose provided that it is properly attributed. For attribution, the original author(s), title, publication source (PeerJ) and either DOI or URL of the article must be cited.
License URL: https://creativecommons.org/licenses/by/4.0/

Keywords: Salmonella Infantis, Whole-genome sequencing, Comparative genomics, Pangenome analysis, wgMLST, Gegenees, Prophages

Funding: FDA/Center for Food Safety and Applied Nutrition (CFSAN) São Paulo Research Foundation (FAPESP) 2019/19338-8 2019/06947-6 National Council for Scientific and Technological Development (CNPq) Proc. 141017/2021-0 Proc. 304399/2018-3 Proc. 304803/2021-9 Coordenação de Aperfeiçoamento de Pessoal de Nível Superior—Brasil (CAPES)—Finance 001 This study was supported by research grants from the FDA/Center for Food Safety and Applied Nutrition (CFSAN) under the supervision of Marc W Allard and from the São Paulo Research Foundation (FAPESP; Proc. 2019/19338-8) under the supervision of Juliana P. Falcão. During the course of this work, Felipe P. Vilela was supported by Master and PhD student scholarships from FAPESP (Proc. 2019/06947-6) and the National Council for Scientific and Technological Development (CNPq; Proc. 141017/2021-0), respectively. Juliana P. Falcão also received Productive fellowships from CNPq (Proc. 304399/2018-3 and 304803/2021-9). This study was financed by the Coordenação de Aperfeiçoamento de Pessoal de Nível Superior—Brasil (CAPES)—Finance Code 001. The funders had no role in study design, data collection and analysis, decision to publish, or preparation of the manuscript.

==============================
Background

Salmonella enterica serovar Infantis (Salmonella Infantis) is a zoonotic, ubiquitous and foodborne pathogen of worldwide distribution. Despite Brazil’s relevance as a major meat exporter, few studies were conducted to characterize strains of this serovar by genomic analyses in this country. Therefore, this study aimed to assess the diversity of 80 Salmonella Infantis strains isolated from veterinary, food and human sources in Brazil between 2013 and 2018 by comparative genomic analyses. Additional genomes of non-Brazilian countries (n = 18) were included for comparison purposes in some analyses.

Methods

Analyses of whole-genome multi-locus sequence typing (wgMLST), using PGAdb-builder, and of fragmented genomes, using Gegenees, were conducted to compare the 80 Brazilian strains to the 18 non-Brazilian genomes. Pangenome analyses and calculations were performed for all Salmonella Infantis genomes analyzed. The presence of prophages was determined using PHASTER for the 80 Brazilian strains. The genome plasticity using BLAST Ring Image Generator (BRIG) and gene synteny using Mauve were evaluated for 20 selected Salmonella Infantis genomes from Brazil and ten from non-Brazilian countries. Unique orthologous protein clusters were searched in ten selected Salmonella Infantis genomes from Brazil and ten from non-Brazilian countries.

Results

wgMLST and Gegenees showed a high genomic similarity among some Brazilian Salmonella Infantis genomes, and also the correlation of some clusters with non-Brazilian genomes. Gegenees also showed an overall similarity >91% among all Salmonella Infantis genomes. Pangenome calculations revealed an open pangenome for all Salmonella Infantis subsets analyzed and a high gene content in the core genomes. Fifteen types of prophages were detected among 97.5% of the Brazilian strains. BRIG and Mauve demonstrated a high structural similarity among the Brazilian and non-Brazilian isolates. Unique orthologous protein clusters related to biological processes, molecular functions, and cellular components were detected among Brazilian and non-Brazilian genomes.

Conclusion

The results presented using different genomic approaches emphasized the significant genomic similarity among Brazilian Salmonella Infantis genomes analyzed, suggesting wide distribution of closely related genotypes among diverse sources in Brazil. The data generated contributed to novel information regarding the genomic diversity of Brazilian and non-Brazilian Salmonella Infantis in comparison. The different genetically related subtypes of Salmonella Infantis from Brazil can either occur exclusively within the country, or also in other countries, suggesting that some exportation of the Brazilian genotypes may have already occurred.

Background

Non-typhoidal Salmonella enterica (NTS) serovars are one of the four major bacterial pathogens related to human foodborne diseases worldwide, with estimated 93.8 million gastroenteritis cases and 155 thousand deaths per year (Majowicz et al., 2010; WHO, 2022). Salmonella enterica subspecies enterica serovar Infantis (Salmonella Infantis) is NTS, ubiquitous, zoonotic and globally distributed serovar. Despite its presence in several types of isolation sources, the main reservoirs of Salmonella Infantis are food-producing animals, with a higher prevalence in poultry (Crim et al., 2015; EFSA & ECDC, 2019). In humans, the main clinical manifestation of Salmonella Infantis infection are gastroenteritis cases developed through the consumption of contaminated raw or undercooked meat products. The increasing resistance rates to antimicrobial drugs of clinical and non-clinical use also raised an alert over this serovar, in special due to the global dissemination of the pESI plasmid, that harbours genes related to antimicrobial resistance, heavy metal tolerance, virulence and stress adaptation (Alvarez et al., 2023).

In Brazil, strains of this serovar may be an important issue for the food safety and public health fields. The country currently ranks among the world’s leading meat exporters (ABPA Brazilian Association of Animal Protein, 2021), and previous reports have demonstrated high frequencies of this serovar among food, environmental, animal and human sources (Medeiros et al., 2011; Voss-Rech et al., 2015; Monte et al., 2019; Vilela et al., 2022a).

Over the years, methods such as pulsed-field gel electrophoresis (PFGE) and multi-locus sequence typing (MLST) have been broadly employed for studying NTS serovars, including Salmonella Infantis strains in Brazil (Almeida et al., 2013; Monte et al., 2019; Vilela et al., 2022a). Currently, whole-genome sequencing (WGS) has been providing significant advances in monitoring genomic relationships and antimicrobial resistance development among bacterial pathogens due to the development of novel methods of analysis, broader access and cost reductions (Gilmour et al., 2013; Allard et al., 2018). As a result, WGS is now an essential tool in the global tracking of zoonotic and foodborne pathogens of public health importance, thanks to its discriminative nature, speed, and cost-effectiveness (Gilmour et al., 2013; Allard et al., 2018). Such advances in WGS have been driven by international efforts to integrate human, animal, and environmental health through the One Health philosophy (Gilmour et al., 2013; Allard et al., 2018).

Different phylogenetic approaches can be applied in the analysis of WGS data, such as single-nucleotide polymorphisms (SNPs), gene-by-gene analysis such as core and whole genome MLST, fragmented genomes, circular comparisons, gene synteny and pangenome calculations (Darling, Mau & Perna, 2010; Alikhan et al., 2011; Agren et al., 2012; Liu, Chiou & Chen, 2016). Genetic markers, such as antimicrobial resistance and virulence genes, pathogenicity islands, orthologous proteins, plasmids and prophages, can also be searched using WGS-based tools. In a global context, several published studies have analyzed Salmonella Infantis strains employing WGS-based methods, such as SNP analyses, MLST and core genome MLST. Investigators also have searched for plasmids, antimicrobial resistance and virulence genes (Brown et al., 2018; Acar et al., 2019; Alba et al., 2020; Nagy et al., 2020; Egorova et al., 2021; Kürekci et al., 2021; Szmolka, Wami & Dobrindt, 2021; Tyson et al., 2021; Papić et al., 2022). However, in Brazil, genomic information is still scarce for this serovar (Monte et al., 2019; Melo et al., 2021; Santos et al., 2021; Bertani et al., 2022; Vilela et al., 2022a; Vilela et al., 2022b).

Considering the limited genomic information available about this serovar in Brazil, the aims of this study were to characterize Salmonella Infantis strains isolated from food, farm and industry environments, humans, animals, and animal feed from 2013 to 2018 in Brazil using comparative genomic analyses, and to establish correlations between these strains and isolates from other countries.

Materials and Methods

Brazilian bacterial genomes

In the present study, 80 draft genomes of Salmonella Infantis, isolated from food (n = 27), farm and industry environments (n = 24), humans (n = 19), animals (n = 7) and animal feed (n = 3) between 2013 and 2018 in different states of Brazil were analyzed. The Salmonella reference laboratory of the Oswaldo Cruz Foundation (FIOCRUZ-RJ) provided these strains. Their genomic DNA was extracted through the phenol-chloroform-isoamyl alcohol technique (Vilela et al., 2021). Libraries preparation was conducted with 1ng of genomic DNA using Nextera XT DNA library preparation kit (Illumina, San Diego, CA, USA). A minimum target coverage of 30X was established. Genomes were sequenced in the Illumina MiSeq platform using the 2 × 150-bp paired-end MiSeq reagent kit version 3 (Illumina, San Diego, CA, USA). Genome drafts were assembled using SKESA 2.2 and annotated using NCBI’s Prokaryotic Genome Annotation Pipeline (PGAP). Quality control was performed in MicroRunQC workflow (Vilela et al., 2021).

These 80 genomes have been previously characterized by a whole-genome SNP analyses using NCBI Pathogen Detection (https://www.ncbi.nlm.nih.gov/pathogens/; Vilela et al., 2022a), and 72 of these genomes have been assigned to 15 different SNP clusters. It was demonstrated that strains from nine specific SNP clusters (accession numbers PDS000029248.42; PDS000018462.80; PDS000028532.4; PDS000074308.3; PDS000106150.2; PDS000075101.1; PDS000026846.6; PDS000016779.5; PDS000140572.1) were related to genomes from other countries besides Brazil, while strains assigned to six other SNP clusters (accession numbers PDS000078471.1; PDS000074309.3; PDS000074994.3; PDS000078491.1; PDS000020042.7; PDS000078459.1;) were only related among each other or to other Brazilian genomes.

Accession numbers, SNP clusters, the years, sources, materials and states of isolation of the 80 Salmonella Infantis draft genomes from Brazil are available in Table 1. Detailed information regarding the sequencing of the 80 Salmonella Infantis strains from Brazil, as well as its respective accession numbers, have also been previously reported (Vilela et al., 2021).

Table 1 Metadata of the draft genomes of 98 Salmonella Infantis strains isolated from food (n = 31), environmental (n = 26), human (n = 28), animal (n = 9), and animal feed (n = 4) sources analyzed in this study.

Country±	Strain	Year	Isolation source	Isolation material	GenBank accession	Pathogen Detection SNP Cluster	
Brazil (PR)	SI 1348/13	2013	Human	Human feces	GCA_015005295.1	PDS000029248.42	
Brazil (PR)	SI 2385/13*,#	2013	Food	Soy	GCA_015004955.1	PDS000106150.2	
Brazil (AL)	SI 2950/13	2013	Human	Human feces	GCA_015005655.1	PDS000074308.3	
Brazil (AL)	SI 2951/13	2013	Human	Human feces	GCA_015005535.1	PDS000074308.3	
Brazil (SC)	SI 3156/13	2013	Environment	Disposable shoe cover	GCA_015004835.1	PDS000018462.80	
Brazil (SC)	SI 5025/13*	2013	Human	Human feces	GCA_015004775.1	–	
Brazil (RS)	SI 124/14	2014	Animal	Swine feces	GCA_015000505.1	PDS000074309.3	
Brazil (SC)	SI 210/14	2014	Environment	Dragging swab	GCA_015000845.1	PDS000018462.80	
Brazil (SC)	SI 212/14	2014	Environment	Dragging swab	GCA_015000465.1	PDS000018462.80	
Brazil (SP)	SI 388/14*#	2014	Animal feed	Soybean animal meal	GCA_015000865.1	–	
Brazil (SC)	SI 583/14	2014	Food	Chicken carcass	GCA_015000785.1	PDS000018462.80	
Brazil (SC)	SI 584/14	2014	Food	Pasta containing ham	GCA_015001025.1	PDS000018462.80	
Brazil (SC)	SI 677/14*	2014	Food	Carcass cleaning wipe	GCA_015004135.1	–	
Brazil (SC)	SI 723/14	2014	Environment	Dragging swab	GCA_015003895.1	PDS000018462.80	
Brazil (RS)	SI 982/14	2014	Animal	Chicken feces	GCA_015005755.1	PDS000074309.3	
Brazil (RS)	SI 1143/14*	2014	Animal	Chicken feces	GCA_015005715.1	PDS000074309.3	
Brazil (SC)	SI 1284/14	2014	Environment	Dragging swab	GCA_015006215.1	PDS000018462.80	
Brazil (RS)	SI 1380/14	2014	Animal	Chicken feces	GCA_015005255.1	PDS000074309.3	
Brazil (RS)	SI 1408/14	2014	Human	Human feces	GCA_015005915.1	PDS000029248.42	
Brazil (RS)	SI 1409/14	2014	Human	Human feces	GCA_015005275.1	PDS000029248.42	
Brazil (RS)	SI 1441/14*	2014	Food	Mayonnaise	GCA_015005515.1	PDS000029248.42	
Brazil (RS)	SI 1711/14	2014	Animal	Chicken feces	GCA_016437325.1	PDS000074309.3	
Brazil (SC)	SI 2378/14	2014	Environment	Truck swab	GCA_015005635.1	PDS000018462.80	
Brazil (SC)	SI 2430/14	2014	Food	Mixed meat sausage	GCA_015005595.1	PDS000018462.80	
Brazil (SC)	SI 2461/14	2014	Food	Chicken carcass	GCA_015004875.1	PDS000018462.80	
Brazil (SC)	SI 2463/14	2014	Food	Chicken carcass	GCA_016437365.1	PDS000018462.80	
Brazil (RS)	SI 2548/14	2014	Animal	Chicken feces	GCA_015000545.1	PDS000029248.42	
Brazil (RS)	SI 3836/14*	2014	Environment	Dragging swab	GCA_015243075.1	PDS000075101.1	
Brazil (MG)	SI 4882/14*	2014	Food	Chicken carcass	GCA_015244115.1	PDS000074994.3	
Brazil (MG)	SI 4892/14	2014	Food	Chicken wings	GCA_015223455.1	PDS000074994.3	
Brazil (MG)	SI 4895/14	2014	Food	Chicken carcass	GCA_015221915.1	PDS000074994.3	
Brazil (MG)	SI 4901/14	2014	Food	Pig snout	GCA_015223515.1	PDS000074994.3	
Brazil (MG)	SI 5247/14	2014	Food	Chicken upper leg and thigh	GCA_015221895.1	PDS000074994.3	
Brazil (SC)	SI 342/15	2015	Food	Swine heart	GCA_015527045.1	PDS000018462.80	
Brazil (SC)	SI 444/15	2015	Food	Pork filet	GCA_015526385.1	PDS000018462.80	
Brazil (SC)	SI 447/15	2015	Food	Smoked and salted pork meat	GCA_015525665.1	PDS000018462.80	
Brazil (SC)	SI 1809/15*,#	2015	Animal feed	Meat animal meal	GCA_015526085.1	PDS000018462.80	
Brazil (SC)	SI 1816/15	2015	Animal feed	Poultry viscera animal meal	GCA_015527645.1	PDS000018462.80	
Brazil (SC)	SI 2280/15	2015	Food	Chicken carcass	GCA_015527445.1	PDS000018462.80	
Brazil (SC)	SI 2302/15	2015	Environment	Cleaning wipe	GCA_015527305.1	PDS000029248.42	
Brazil (SC)	SI 2370/15	2015	Food	Carcass cleaning wipe	GCA_015527525.1	PDS000029248.42	
Brazil (MG)	SI 2869/15	2015	Food	Chicken upper leg	GCA_015527575.1	PDS000074994.3	
Brazil (MG)	SI 3056/15	2015	Food	Chicken carcass	GCA_015527355.1	PDS000074994.3	
Brazil (SC)	SI 4764/15	2015	Environment	Cleaning wipe	GCA_015527945.1	PDS000018462.80	
Brazil (SC)	SI 5391/15	2015	Environment	Disposable shoe cover	GCA_015527425.1	PDS000020042.7	
Brazil (SC)	SI 5837/15*,#	2015	Environment	Disposable shoe cover	GCA_015526805.1	PDS000026846.6	
Brazil (SC)	SI 5853/15*,#	2015	Environment	Disposable shoe cover	GCA_015598965.2	PDS000020042.7	
Brazil (SC)	SI 5859/15	2015	Environment	Disposable shoe cover	GCA_015528025.1	PDS000020042.7	
Brazil (SC)	SI 5911/15	2015	Environment	Cleaning wipe	GCA_015527665.1	PDS000018462.80	
Brazil (SC)	SI 5912/15	2015	Environment	Cleaning wipe	GCA_016436965.1	PDS000018462.80	
Brazil (SC)	SI 5915/15	2015	Environment	Cleaning wipe	GCA_016437765.1	PDS000018462.80	
Brazil (SC)	SI 5923/15	2015	Environment	Cleaning wipe	GCA_016437485.1	PDS000018462.80	
Brazil (SC)	SI 220/16	2016	Environment	Cleaning wipe	GCA_016437465.1	PDS000018462.80	
Brazil (SC)	SI 3687/16	2016	Food	Chicken carcass	GCA_016437445.1	PDS000018462.80	
Brazil (SC)	SI 4447/16	2016	Food	Pork sausage	GCA_016437565.1	PDS000018462.80	
Brazil (SC)	SI 5946/16	2016	Food	Pork rib	GCA_016211625.1	PDS000018462.80	
Brazil (MA)	SI 6987/16*,#	2016	Human	Human feces	GCA_016230965.1	PDS000140572.1	
Brazil (RS)	SI 7876/16	2016	Human	Human feces	GCA_016220265.1	PDS000018462.80	
Brazil (PR)	SI 11/17*,#	2017	Environment	Dragging swab	GCA_016224725.1	PDS000028532.4	
Brazil (PR)	SI 23/17	2017	Environment	Dragging swab	GCA_016222525.1	PDS000028532.4	
Brazil (PR)	SI 238/17	2017	Environment	Dragging swab	GCA_016222645.1	PDS000029248.42	
Brazil (MG)	SI 872/17	2017	Food	Chicken carcass	GCA_016220425.1	PDS000078491.1	
Brazil (SP)	SI 1171/17*	2017	Environment	Soil	GCA_016217285.1	–	
Brazil (SP)	SI 1256/17*,#	2017	Environment	Soil	GCA_016222625.1	–	
Brazil (SC)	SI 2580/17*	2017	Human	Human feces	GCA_016438385.1	PDS000016779.5	
Brazil (GO)	SI 2953/17	2017	Human	Human fecal swab	GCA_016437405.1	PDS000078459.1	
Brazil (GO)	SI 2954/17	2017	Human	Human fecal swab	GCA_016437305.1	PDS000078459.1	
Brazil (GO)	SI 3380/17*	2017	Human	Human fecal swab	GCA_016438725.1	PDS000078459.1	
Brazil (MG)	SI 3877/17	2017	Food	Chicken wings	GCA_016437385.1	PDS000078491.1	
Brazil (SP)	SI 3906/17	2017	Environment	Sieve residue	GCA_016437045.1	PDS000018462.80	
Brazil (PR)	SI 4065/17	2017	Human	Human feces	GCA_016436945.1	PDS000018462.80	
Brazil (PR)	SI 4067/17	2017	Human	Human feces	GCA_016437525.1	PDS000018462.80	
Brazil (PR)	SI 4069/17	2017	Human	Human blood	GCA_016437225.1	PDS000074309.3	
Brazil (MG)	SI 52/18	2018	Food	Chicken carcass	GCA_016437345.1	PDS000078491.1	
Brazil (GO)	SI 331/18*	2018	Human	Human fecal swab	GCA_016436865.1	–	
Brazil (SC)	SI 623/18	2018	Human	Human feces	GCA_016437505.1	PDS000029248.42	
Brazil (MS)	SI 661/18*,#	2018	Human	Human feces	GCA_016438625.1	PDS000078471.1	
Brazil (RS)	SI 942/18	2018	Human	Human fecal swab	GCA_016437265.1	PDS000018462.80	
Brazil (SC)	SI 1634/18	2018	Food	Yellowtail amberjack fish meat	GCA_016436825.1	PDS000029248.42	
Brazil (GO)	SI 2676/18*,#	2018	Animal	Avian reproductive matrix	GCA_016437245.1	PDS000078491.1	
Australia	AUSMDU00032459*,#	2019	Human	Clinical	GCA_032723255.1	PDS000027076.745	
Bolivia	SBO17	2015	Environmental	Hospital	GCA_012223145.1	PDS000029248.42	
Germany	15-SA02526*,#	2015	Food	Food	GCA_010615385.1	PDS000032463.111	
Germany	17-SA00182*,#	2016	Animal	Animal	GCA_010614605.1	PDS000003948.23	
Hungary	SIB16*,#	2012	Animal	Broiler feces	GCA_016944915.1	PDS000077501.33	
Israel	119944*,#	2008	Human	Stool	GCA_000506925.1	PDS000032399.9	
Mexico	MPSPSA2193-1*,#	2022	Environmental	Dam	GCA_032351395.1	PDS000108885.76	
Paraguay	ERS13618234	2020	Human	Blood	GCA_026305245.1	PDS000018462.80	
Peru	3.591-2010*,#	2010	Food	Meat	GCA_005970285.1	PDS000089910.508	
Russia	VGNKI000011*,#	2017	Animal feed	Chicken feed	GCA_008361015.1	–	
South Africa	741581*,#	2013	Human	Stool	GCA_024369075.1	PDS000032399.9	
Turkey	MET-S1-498*,#	2012	Food	Chicken meat	GCA_010937355.1	PDS000091376.17	
United Kingdom	528502	2018	Food	Food	GCA_007269555.1	PDS000026846.6	
United Kingdom	279940	2016	Human	Human	GCA_009520195.1	PDS000016779.5	
United Kingdom	hPHE_193	2013	Human	Human	GCA_018844535.1	PDS000074308.3	
United States	PNUSAS339754	2023	Human	Human	GCA_029442055.1	PDS000140572.1	
United States	PNUSAS208452	2021	Human	Human	GCA_019046105.1	PDS000028532.4	
United States	PNUSAS082678	2019	Human	Human	GCA_007620775.1	PDS000075101.1	
Notes.

± The codes between parenthesis in the strains from Brazil represent the following states of the country: RS, Rio Grande do Sul; PR, Paraná; SC, Santa Catarina; SP, São Paulo; MG, Minas Gerais; MS, Mato Grosso do Sul; GO, Goiás; BA, Bahia; AL, Alagoas; PE, Pernambuco; MA, Maranhão.

* Strains selected for the analysis of genome plasticity using Blast Ring Image Generator and gene synteny using Mauve.

# Strains selected for the search of orthologous protein clusters using OrthoVenn2.

Data regarding the 80 Brazilian genomes are available in Vilela et al. (2021).

Additional non-Brazilian genomes

For comparison purposes, 18 representative draft genomes of Salmonella Infantis strains from countries other than Brazil that were publicly available for download at NCBI Pathogen Detection were included in some analyses performed (Table 1). These genomes came from strains isolated in Australia (n = 1), Bolivia (n = 1), Germany (n = 2), Hungary (n = 1), Israel (n = 1), Mexico (n = 1), Paraguay (n = 1), Peru (n = 1), Russia (n = 1), South Africa (n = 1), Turkey (n = 1), the United Kingdom (UK; n = 3), and the United States (USA; n = 3), from human, food, environmental and animal sources.

A total of 18 additional Salmonella Infantis draft genomes from other countries were included as representative samples: eight draft genomes from SNP clusters common to the ones of the Brazilian genomes (item 2.1; Vilela et al., 2022a; Table 1); eight draft genomes from other SNP clusters then the ones described in item 2.1, representing the most frequently detected SNP clusters at NCBI Pathogen Detection; and two draft genomes of different SNP clusters carrying the epidemic pESI plasmid.

The closed genome of Salmonella Infantis reference strain SINFA (GenBank accession number LN649235.1) was included either for comparison purposes or as reference for alignments where mentioned. Strain SINFA was isolated in the United Kingdom in 1973 from chicken. The closed genome of Escherichia coli (E. coli) reference strain K-12 (GenBank accession number NC_000913.3) was included as an outgroup where mentioned.

Accession numbers, SNP clusters, years, sources, materials and countries of isolation of the draft genomes of the 18 non-Brazilian Salmonella Infantis draft genomes included for comparison are available in Table 1.

Phylogenomic relationship and similarity analysis

Analyses of wgMLST and fragmented genomes were performed to determine the phylogenomic relationship among the 80 Brazilian and 18 non-Brazilian Salmonella Infantis strains (Table 1), plus references Salmonella Infantis SINFA and E. coli K-12.

The wgMLST analysis was performed using the PGAdb-builder web service (http://wgmlstdb.imst.nsysu.edu.tw/; Liu, Chiou & Chen, 2016). Initially, in the Build_PGAdb module, all genomic files (limited to 100 genomes per analysis by the website) were uploaded and analyzed for the determination of its allelic profile. After the creation of the allelic profile, all genomic files were re-uploaded in the Build_wgMLSTtree module of the same website in order to construct a phylogenetic tree by the BLASTn method, using 90% of coverage and identity as parameters for the alignment. The specifics of PGAdb-builder web service, including the Build_PGAdb and Build_wgMLSTtree modules, have been previously described (Liu, Chiou & Chen, 2016). The resulting phylogenetic tree in the newick extension was edited with FigTree v. 1.4.4 software (http://tree.bio.ed.ac.uk/software/figtree/). A second wgMLST analysis was conducted using the same parameters, only with the 80 Salmonella Infantis genomes from Brazil, in order to determine possible correlations with the profiles of antimicrobial resistance, efflux pump encoding and heavy metal tolerance genes previously described (Vilela et al., 2022a; Vilela et al., 2022b).

The analysis of fragmented genomes was conducted in the software Gegenees 3.1 (Agren et al., 2012). Genomic files were imported to the software and fragmented using a 500 bp fragmentation length and step size. The fragments were aligned and compared using the BLASTn method, and similarity scores were calculated. The resulting similarity matrix was exported to the SplitsTree4 software (Huson & Bryant, 2006) to create a phylogenetic tree based on the Neighbor-Joining method. The specifics of Gegenees have been previously described (Agren et al., 2012).

Pangenome analysis

The pangenome analysis was performed using the amino-acid sequences of the draft genomes, containing all coding DNA sequences (CDSs). Five different subsets were analyzed: (A) the 80 Brazilian Salmonella Infantis; (B) the eight non-Brazilian Salmonella Infantis with the same SNP clusters assigned to the Brazilian genomes; (C) the 10 non-Brazilian Salmonella Infantis of SNP clusters different than those assigned to the Brazilian genomes; (D) the 18 non-Brazilian Salmonella Infantis combined; and (E) the total 98 Salmonella Infantis analyzed (18 non-Brazilian and 80 Brazilian; Table 1).

Amino-acid sequences of all CDSs of the genomes were analyzed by the OrthoFinder software (Emms & Kelly, 2015) to predict orthologous gene clusters through an all-vs-all Diamond method. Then, clusters were grouped with the Markov clustering (MCL) algorithm (Enright, Van Dongen & Ouzounis, 2002). The results obtained were used to classify the gene clusters into three groups: core genome (genes present in all strains), accessory genome (genes present in some, but not all strains) and singletons (genes present in only one strain). This classification was performed based on a previously described in-house script (Soares et al., 2013). The extrapolation of the pangenome was calculated by curve fitting based on Heap’s Law. Values ≤ 1 indicate an open pangenome, where each added genome may contribute with some new genes, which would increase the pangenome. Values >1 indicate a closed pangenome, where the addition of new genomes will not significantly affect the size of the pangenome. The extrapolations of the core genomes and singletons were calculated by curve fitting based on least-squares fit of the exponential regression decay. The formulas and calculation of the extrapolation of the pangenome, core genome and singletons were performed as previously described (Benevides et al., 2017).

Search of prophages

Prophages were searched in the 80 Brazilian Salmonella Infantis draft genomes analyzed (Table 1) using the web service PHASTER (https://phaster.ca/; Arndt et al., 2016). Genomes were individually uploaded in the website and default parameters were applied. In the summary of results obtained, only prophages with intact sequences (score >90) were included in the results reported here. The specifics of PHASTER have been previously described (Arndt et al., 2016).

Genome plasticity and gene synteny

Genome plasticity and gene synteny were evaluated for 20 Salmonella Infantis genomes from Brazil and 10 non-Brazilian genomes (marked with an asterisk (*) in Table 1), that were selected based on the different SNP clusters included in the present study.

Genome plasticity was evaluated using BRIG (Alikhan et al., 2011). The 30 draft genomes and the complete genome of Salmonella Infantis SINFA (used as a reference for the alignment) were imported into the software and aligned by the BLASTn method. Results were plotted in a circular genome map composed of multiple rings of different color, each corresponding to one of the genomes included in the analysis. The blank segments in the rings indicate deleted areas in the strains analyzed in comparison to the reference genome used. The specifics of BRIG have been previously described (Alikhan et al., 2011).

Gene synteny analysis was conducted using the Mauve (Darling, Mau & Perna, 2010). The 30 draft genomes and the complete genome of Salmonella Infantis SINFA were analyzed using the “progressiveMauve” algorithm with default parameters. The specifics of Mauve have been previously described (Darling, Mau & Perna, 2010).

Search for orthologous proteins

The search for unique clusters of orthologous proteins was performed for the Salmonella Infantis reference strain SINFA in comparison to three distinct subsets: (A) ten Brazilian genomes that were selected based on the profiles observed in the BRIG analysis; (B) the eight non-Brazilian Salmonella Infantis genomes with SNP clusters common to the Brazilian genomes; (C) the eight non-Brazilian Salmonella Infantis genomes of SNP clusters different than those assigned to the Brazilian genomes; and (D) the two pESI-positive genomes from Israel and Russia (Table 1). The genomes selected for this analysis are marked with # in Table 1. The search was performed using the web service OrthoVenn3 (https://orthovenn3.bioinfotoolkits.net/). Amino-acid sequences were submitted in the “Start” field using default parameters. In the output generated, only protein clusters present in all strains studied and absent in the SINFA reference genome were included as results. The specifics of OrthoVenn3 have been previously described (Sun et al., 2023).

Results

wgMLST and Gegenees

The phylogenomic trees generated by wgMLST and Gegenees analyses are displayed, respectively, in Figs. 1 and 2. Both methods showed, in overall, a separation among Brazilian and some non-Brazilian genomes.

Figure 1 wgMLST phylogenomic tree generated with PGAdb-builder with 80 Brazilian and 18 non-Brazilian Salmonella Infantis draft genomes.

Sources: food (FO; red), environmental (EN; green), human (HU; blue), animal (AN; yellow), animal feed (RF; orange). Reference genomes (RF; black): Salmonella Infantis SINFA and Escherichia coli K-12.

Figure 2 Fragmented genome phylogenetic tree based on the Gegenees analysis of 80 Brazilian and 18 non-Brazilian Salmonella Infantis draft genomes.

The differences in the level of similarity found among the strains analyzed are represented by green shades (higher percentages of similarities) to red shades (smaller similarity percentages). Similarity percentages are also demonstrated by the numbers inside each square, that represents the similarity percentage among two strains. Sources: food (FO; red), environmental (EN; green), human (HU; blue), animal (AN; yellow), animal feed (RF; orange). Reference genomes (RF; black): Salmonella Infantis SINFA and Escherichia coli K-12.

In wgMLST, five major groups were formed. The two largest groups comprised 72 Brazilian genomes of strains isolated from food, environment, human and animal feed, and the eight non-Brazilian genomes with common SNP clusters, from USA, UK, Bolivia and Paraguay. The remaining seven genomes from Brazil of strains isolated from animals and a single environmental strain were located in a separate group. The 10 non-Brazilian genomes that were assigned to different SNP clusters from those assigned for Brazilian genomes were grouped into two different groups: one composed of genomes from Australia, Mexico and Salmonella Infantis reference SINFA, and another with genomes from Germany, Hungary, Israel, Peru, Russia, South Africa and Turkey (Fig. 1).

Gegenees also showed five main groups, but with some differences in relation to wgMLST (Fig. 2). All 80 genomes from Brazil were distributed in four major groups, with no clear separation regarding the strains’ isolation sources. In addition, the genomes from Australia and Mexico with different SNP clusters and four genomes from Bolivia, Paraguay, UK and USA with common SNP clusters were also grouped along with Brazilian genomes in these same groups. The fifth group formed was comprised by the eight genomes from Germany, Hungary, Israel, Peru, Russia, South Africa and Turkey of different SNP clusters and four genomes from the UK and the USA from common SNP clusters to Brazilian genomes (Fig. 2). Gegenees also showed an overall similarity >91% among all Salmonella Infantis genomes included in the comparison (Fig. 2).

In the wgMLST analysis conducted to evaluate the genomic relationship of the antimicrobial resistance gene profiles of the 80 Brazilian genomes (Fig. S1), it was possible to observe the occurrence of three major profiles/groups. The majority of the strains of Profile 1 were characterized by the presence of genes encoding aminoglycoside resistance (aac(6′)-Iaa), multi-drug efflux pumps (mdsA, mdsB, acrA, acrB, baeR, crp, emrB, emrR, golS, hns, kdpE, kpnF, marA, marR, mdfA, mdtK, msbA, rsmA, sdiA, soxR, soxS), arsenic tolerance (arsR), and gold tolerance (golS and golT). The strains of Profile 2 shared most of the genes of Profile 1, but differed in the absence of mdfA and in the presence of genes encoding resistance to beta-lactams (bla TEM−1), phenicols (floR), trimethoprim (dfrA8), tetracycline (tet(A)) and silver tolerance (silABCDEFPRS). The strains of Profile 3 differed of Profile 1 in the presence of beta-lactam resistance gene bla CMY −2.

Pangenome analysis

Figure 3 shows the number of pangenome CDSs, the pangenome development calculated using Heap’s law (considering α = 1- γ), and the accessory genome and singletons development, calculated by curve fitting based on least-squares fit of the exponential regression decay, for the five Salmonella Infantis subsets analyzed.

Figure 3 The number of coding DNA sequences (CDSs) on the pangenome subsets contributing to the development of the pangenome, accessory genome and singletons of the Salmonella Infantis.

(A) Eighty Brazilian Salmonella Infantis draft genomes. (B) Eighteen non-Brazilian Salmonella Infantis draft genomes. (C) Ten non-Brazilian Salmonella Infantis draft genomes of different SNP clusters (dSNP) than the 80 Brazilian genomes. (D) Eight non-Brazilian Salmonella Infantis draft genomes of the same SNP clusters (dSNP) of the 80 Brazilian genomes. (E) Eighteen non-Brazilian and 80 Brazilian (n = 98) Salmonella Infantis draft genomes.

All five Salmonella Infantis subsets analyzed showed the presence of an open pangenome (subset A, α = 0.92; subset B, α = 0.907; subset C, α = 0.936; subset D, α = 0.958; subset E, α = 0.925). The pangenome of subset A (Brazilian genomes) showed the highest proportion and number of genes in the core genome, with 4,066 CDSs identified. The pangenome of subset D (non-Brazilian genomes with common SNP clusters to Brazilian strains) showed the highest proportion and number of genes in the accessory genome, with 668 CDSs identified. The pangenome of subset C (non-Brazilian genomes with different SNP than Brazilian strains) showed the highest proportion and number of singletons, with 209 CDSs identified (Fig. 3).

Frequency and diversity of prophages

A total of 78 (97.5%) of the 80 Brazilian Salmonella Infantis genomes analyzed, harbored from one up to four prophages of the 15 types detected in this study (Table S1). Prophage Salmon Fels 1 was detected in 39 strains (48.75%), Gifsy 1 in 32 (40%), Entero BP 4795 and Salmon 118970 sal3 in 28 (35%), Yersin L 413C in 16 (20%), Salmon SPN3UB in eight (10%), Escher pro483 in seven (8.75%), Stx2 c 1717 and Salmon SW9 in four (5%), Entero P4 and Salmon vB SosS Oslo in three (3.75%), and Entero ES18, Entero YYZ 2008, Entero fiAA91 ss and Salmon SP 004 in single strains (1.25%). Only strains 1171/17 and 1256/17 did not harbor any prophages.

Genome plasticity and gene synteny

In the circular map generated with BRIG for the evaluation of genome plasticity (Fig. 4), it is possible to observe few conserved deletion fields (blank spaces) among the 20 Brazilian and ten non-Brazilian Salmonella Infantis analyzed, aligned with Salmonella Infantis reference genome SINFA.

Figure 4 Genomic plasticity of 20 Brazilian and ten non-Brazilian Salmonella Infantis draft genomes created with the BRIG software using the Salmonella Infantis strain SINFA as a reference.

Each ring in the image corresponds to one of the Salmonella Infantis draft genomes analyzed, with the respective corresponding color and number to the legend at the right. Deletion regions are represented by blank spaces inside the rings, while shared regions are filled with color. The 30 genome rings were manually numbered from 1 to 30 using Microsoft Paint 3D for better visualization.

In the gene synteny analysis of 20 Brazilian and ten non-Brazilian Salmonella Infantis draft genomes, it is possible to observe the presence of large, conserved and less numerous Locally Collinear Blocks (LCBs), with few areas of deletions, inclusions and inversions, in comparison to the Salmonella Infantis reference SINFA (Fig. 5).

Figure 5 Gene synteny analysis of 20 Brazilian and ten non-Brazilian Salmonella Infantis draft genomes using the Mauve software.

The different locally collinear blocks (LCBs) are represented by different colors, while deletion regions are represented by blank spaces between LCBs. A total of 25 LCBs that were more conserved among the genomes analyzed were manually numbered (using Microsoft Paint 3D) for better identification. The enumerated ruler above the LCBs represents the genomic position.

Unique orthologous protein clusters

All ten Brazilian and 18 non-Brazilian Salmonella Infantis genomes analyzed possessed unique clusters of orthologous proteins related to diverse biological processes, molecular functions, and cellular components in comparison to Salmonella Infantis reference SINFA. Detailed information into the unique clusters of orthologous proteins detected is available in Table 2.

Table 2 Presence of unique orthologous protein clusters among the Brazilian (n = 10) and non-Brazilian (n = 18) genomes in comparison to Salmonella Infantis reference strain SINFA LN649235.1.

Salmonella Infantis genomes	Biological processes	Cellular components	Molecular function	
Brazilian genomes (n = 10)	Cellular response to DNA damage stimulus, cellular response to manganese ion, DNA-templated regulation of transcription, leucine biosynthetic process, nicotinamide riboside transport, oxidation–reduction process, pathogenesis, protein transport, regulation of ATPase activity, regulation of translation, response to antibiotic, response to toxic substances, SOS response, transcriptional attenuation by ribosome, translation, tryptophan biosynthetic process, valine biosynthetic process, viral genome integration into host DNA	Cytoplasm	DNA binding, translation elongation factor activity	
sSNP non-Brazilian genomes (n = 8)	Cellular response to DNA damage stimulus, cellular response to manganese ion, DNA-templated regulation of transcription, leucine biosynthetic process, oxidation–reduction process, pathogenesis, protein transport, regulation of ATPase activity, regulation of translation, response to antibiotic, response to toxic substances, SOS response, transcriptional attenuation by ribosome, translation, tryptophan biosynthetic process, valine biosynthetic process	Cytoplasm	DNA binding	
dSNP non-Brazilian genomes (n = 8)	Cellular response to DNA damage stimulus, DNA-mediated transposition DNA-templated regulation of transcription, leucine biosynthetic process, oxidation–reduction process, pathogenesis, protein transport, regulation of ATPase activity, regulation of translation, rescue of stalled ribosome, response to toxic substance, SOS response, translation	–	DNA binding	
pESI non-Brazilian genomes (n = 2)	Amino acid transport, cellular response to DNA damage stimulus, cytochrome complex assembly, DNA repair, DNA replication, DNA-templated regulation of transcription, oxidation–reduction process, pathogenesis, pillus organization, plasmid maintenance, plasmid partitioning, protein secretion, protein transport, response to antibiotic, response to mercury ion, response to radiation, response to toxic substances, sodium ion transport, SOS response, translation, transmembrane transport	Integral component of membrane, periplasmic space, plasma membrane,	ATP binding, mercury ion transmembrane transporter activity, metal ion binding, oxidoreductase activity	
Notes.

sSNP eight non-Brazilian Salmonella Infantis genomes of SNP clusters common to the 80 Brazilian Salmonella Infantis strains analyzed

dSNP eight non-Brazilian Salmonella Infantis genomes of SNP clusters different than the 80 Brazilian Salmonella Infantis strains analyzed

Discussion

Salmonella Infantis is a NTS, ubiquitous and zoonotic serovar, with widespread global distribution over different isolation sources, and mainly associated with gastroenteritis in humans through the consumption of raw or undercooked poultry meat (Crim et al., 2015; EuropeanFoodSafetyAuthority, 2019). Considering Brazil’s relevance in the global meat exportation market (ABPA Brazilian Association of Animal Protein, 2021), the use of genomic tools and analyses could greatly improve the understanding of specific traits of Salmonella Infantis strains in this country. In this work, different genomic approaches were employed in order to expand the characterization of strains of this serovar in the country.

In the present study, it was possible to observe a closer relationship of Brazilian and non-Brazilian Salmonella Infantis strains of the same SNP clusters assigned by NCBI Pathogen Detection in comparison to non-Brazilian genomes of different SNP clusters. Previously, we have also demonstrated this clear genomic distinction among the same 80 Salmonella Infantis genomes from Brazil analyzed here in comparison to other 40 additional genomes from several countries and distinct SNP clusters using cgMLST (Vilela et al., 2022b).

Regarding the distribution of the Brazilian genomes of Salmonella Infantis observed in wgMLST and Gegenees phylogenies, no clear correlation was observed between any specific lineage and the isolation source or date in the majority of the groups formed by both methods. (Figs. 1 and 2). However, wgMLST interestingly clustered all Salmonella Infantis genomes of strains isolated from animals in a group apart from Brazilian genomes of other sources (Fig. 1), which could indicate the circulation of this specific genomic profile mostly among animals in the country.

When wgMLST was combined to the genotypic antimicrobial resistance profile of the 80 Brazilian strains, it revealed the formation of the three distinct genetic groups showing different resistance gene profiles (Fig. S1). While Profiles 1 and 3 grouped strains predicted to be less resistant, Profile 2 contained strains with genetic markers predicting multi-drug resistance.

In comparison to NCBI Pathogen Detection, while wgMLST accurately clustered Brazilian and non-Brazilian Salmonella Infantis genomes with common SNP clusters in comparison to non-Brazilian genomes of different SNP clusters (Fig. 1), Gegenees produced similar results but with less resolution (Fig. 2). In this analysis, a high overall relatedness was also observed among all Salmonella Infantis genomes (>91%; Fig. 2). While wgMLST employs a gene-by-gene approach, Gegenees analysis is based on the similarities found through the alignment of fragments of entire genomes. Eventual differences in the genome sizes and presence of duplicated genes, genomic islands and repetitive regions, for example, could result in small biases in the Gegenees analysis that would not allow the identification of small nucleotide variations, detected by wgMLST. However, as demonstrated in the present study, although with lower resolution, Gegenees was able to generate some similar results to wgMLST and the SNP clusters defined by NCBI Pathogen Detection.

These results indicated that the majority of the Salmonella Infantis strains from Brazil studied were allocated into different genomic groups comprising strains of diverse origins, suggesting that these might be widespread among human and non-human sources and sharing correlated profiles of antimicrobial resistance genes. Moreover, the results demonstrated genetically related groups comprising Brazilian Salmonella Infantis and non-Brazilian genomes from the United Kingdom and North and South America, suggesting their global circulation, while other profiles are distinct and present exclusively within the country.

In the genome plasticity and gene synteny analyses conducted herein by BRIG and Mauve, respectively, Brazilian and non-Brazilian Salmonella Infantis strains shared a highly conserved genomic structure (Figs. 4 and 5). In the genome plasticity analysis, the selected Salmonella Infantis genomes possessed few conserved deletion areas when aligned to the reference Salmonella Infantis genome SINFA (Fig. 5). Similarly, the gene synteny analysis produced by the Mauve also revealed that the selected Salmonella Infantis genomes analyzed possessed highly similar LCBs and few areas of deletions, inclusions and inversions among each other in comparison to Salmonella Infantis SINFA (Fig. 5).

It is important to address that an equal linear organization of LCBs was not observed for the Brazilian and non-Brazilian genomes in comparison to the reference genome SINFA used in the Mauve analysis (Fig. 5). This characteristic does not necessarily indicate a real change in the genomic ordering of the LCBs. Since the genomes analyzed were in a draft configuration, their genomic sequences do not start in the common expected replication origin as in closed genomes, such as the reference strain SINFA. However, despite the visual difference, the results still corroborate a high similarity observed in the other analyses performed.

Several prophages were found among the 80 Brazilian Salmonella Infantis genomes analyzed (Table S1). In NTS, prophages have been demonstrated to be indicatives of genetic diversity and to contribute to bacterial virulence (Gao et al., 2020). In this study, it is worthy to highlight Fels 1 and Gifsy 1, which are well described phages in NTS and were detected in more than 40% of the strains analyzed. Fels 1 has been reported to act in the production of neuraminidase and superoxide dismutase, while Gifsy 1 has been demonstrated to increase the virulence of Salmonella Typhimurium in mice (Brüssow, Canchaya & Hardt, 2004). Entero BP 4795 was found in 35% of the strains and is related to the production of the Shiga toxin in enterohemorragic Escherichia coli (Creuzburg et al., 2005). Yersin L 413C, detected in 20% of the strains, is a common serotype marker of Yersinia pestis (Garcia et al., 2008). However, little is known about the roles in NTS serovars of other phages detected in the present study, such as Salmon 118970 sal3, Salmon SPN3UB, Escher pro483, Stx2 c 1717, Salmon SW9, Entero P4, Salmon vB SosS Oslo, Entero ES18, Entero YYZ 2008, Entero fiAA91 ss and Salmon SP 004.

Prophages have been previously reported in Salmonella Infantis genomes by Gymoese et al. (2019), whose analysis included isolates from humans and several types of non-human sources from Denmark between 2002 and 2012, and in Pardo-Esté et al. (2021), where strains isolated from chicken producing facilities from Chile between 2018 and 2019 were analyzed. In both studies, a greater diversity of phages was detected in comparison to the results obtained here, including differences in the percentages of phages more commonly found, such as Gifsy 1. Therefore, these results demonstrated that important phages that may contribute in the virulence of NTS were detected in high frequencies among Salmonella Infantis strains from Brazil. In addition, they highlight the necessity for further studies to better investigate the roles of other diverse prophages in Salmonella Infantis and other NTS serovars.

The calculation of the pangenome for NTS and other bacteria has an important role for epidemiological and evolutionary purposes, and even as an important tool for secondary analysis, such as the search in silico for vaccine targets (Benevides et al., 2017; Seribelli et al., 2020; Felice et al., 2021). NTS has been considered as a recombinant bacterial genus with an open pangenome and a growing number of orthologous genes as new genomes are sequenced (Alikhan et al., 2018). In the present study, the five subsets analyzed were confirmed to have an open pangenome, with α values smaller than 1 detected for all comparisons.

Despite of the presence of correlated groups observed in wgMLST (Fig. 1), the high similarity percentages found with Gegenees (Fig. 2) and the close α values obtained in the pangenome calculation, it is interesting to notice how the number of CDSs in the core genome, accessory genome and singletons varied in the five comparisons performed (Fig. 3). For example, the subset of Brazilian genomes showed the highest number of CDSs in their core genome, even when compared to the subset of genomes from the same SNP clusters, which in contrast, showed the biggest accessory genome of all the comparisons (Fig. 3). These results demonstrated that Brazilian isolates of Salmonella Infantis presented a high genomic relatedness in the pangenome analysis, with a high core genome content, which significantly differed from genomes of other countries from common or different SNP clusters. This level of genomic conservation of their core genome could also represent an advantage for the possible development of vaccine targets suited for Salmonella Infantis circulating among animals and humans in Brazil.

The pangenome analysis was employed by Mattock et al. (2022), that characterized 395 genomes isolated from humans from South Africa between 2004 and 2020, and also by the report of Pardo-Esté et al. (2021) mentioned above. While Mattock et al. reported 3,983 genes belonging to the core content in the pangenome, a very similar result to the values obtained here was found. Pardo-Esté et al. (2021) reported a core content of 2,618 genes, which differed from our results. It must also be noticed that both studies have reported much higher values for the accessory genome and singleton genes, demonstrating a greater genomic diversity in comparison to our results. These differences demonstrate how strains from the same serovar but from distinct locations may present different genomic traits, reinforcing the importance of more WGS-based works to unravel the epidemiology of important pathogens such as Salmonella Infantis.

Brazilian and non-Brazilian Salmonella Infantis genomes were also demonstrated to have similar unique orthologous protein clusters related to biological processes, molecular functions and cellular components in relation to Salmonella Infantis reference SINFA. Since SINFA was isolated from chicken in 1973 in the United Kingdom (data available at NCBI’s GenBank BioSample page; accession number SAMEA3106395), these results showed that strains of this serovar, in overall, have evolved in order to acquire novel proteins capable to favor the virulence, survival and adaptation of Salmonella Infantis within its hosts. For example, the most diverse unique orthologous protein clusters observed were those found in the two non-Brazilian genomes positive for the pESI global widespread plasmid (119944 from Israel and VGNKI000011 from Russia). These genomes shared unique orthologous protein clusters related to plasmid maintenance, plasmid partitioning, response to mercury ion and mercury ion transmembrane transporter activity, corroborating to the features already described for Salmonella Infantis harboring these plasmids. Also, such clusters of orthologous protein were not present among pESI-negative genomes from Brazil and other countries, reinforcing the hypothesis that strains of this serovar are still in constant acquisition of novel genetic features.

Finally, it is important to mention some of the limitations of the present study. We must state that the Brazilian sampling of Salmonella Infantis does not comprise an even distribution of isolation sources, years and Brazilian states. Therefore, despite of the corroboration of the results here presented, some groups of strains may be over-represented. The number of non-Brazilian genomes included for comparison purposes (n = 18) was limited when compared to the number of Salmonella Infantis already sequenced. By the date of the present study (August 11th, 2023), more than 27 thousand Salmonella Infantis genomes had been deposited in NCBI Pathogen Detection. This specific number of genomes was selected due to limitations of the platforms employed, and since the main objective was to promote a characterization of Brazilian genomes, the amount of non-Brazilian genomes had to be reduced using specific criteria as mentioned before. Finally, the inability to include the allele differences in the wgMLST tree constructed through PGAdb-builder also complicated the inference of the genetic proximity among the strains within each cluster and between clusters.

Conclusion

The results presented using different genomic approaches emphasized the significant genomic similarity among Brazilian Salmonella Infantis genomes analyzed, suggesting wide distribution of closely related genotypes among diverse sources in Brazil. The data generated contributed to novel information regarding the genomic diversity of Brazilian and non-Brazilian Salmonella Infantis in comparison, indicating that the different genetically related subtypes of Salmonella Infantis from Brazil can either occur exclusively within the country, or also in other countries, suggesting that some exportation of the Brazilian genotypes may have already occurred.

Supplemental Information

Supplemental Information 1 wgMLST phylogenomic tree generated with PGAdb-builder and profiles of antimicrobial resistance genes of 80 Brazilian Salmonella Infantis draft genomes

Sources: food (red), environmental (green), human (blue), animal (yellow), animal feed (orange). Reference genomes (black): Salmonella Infantis SINFA and Escherichia coli K-12. The profiles of antimicrobial resistance genes informed were described previously in (Vilela et al., 2022a) (Journal of Applied Microbiology 132:3327-3342 DOI: 10.1111/jam.15430) and (Vilela et al., 2022b) (PLOS ONE 17:e0277979 DOI: 10.1371/journal.pone.0277979).

Supplemental Information 2 Occurrence of prophages in the 80 Brazilian genomes of Salmonella Infantis strains analyzed in the present study

We thank to the Kentucky Division of Lab Services, Centralized Lab Facility (Frankfort, KY, USA) for performing the whole-genome sequencing of the strains studied, and for Maria Balkey from FDA/CFSAN for the support during this study.

Additional Information and Declarations

Competing Interests

Author Contributions

Data Availability

The authors declare there are no competing interests.

Felipe P. Vilela conceived and designed the experiments, performed the experiments, analyzed the data, prepared figures and/or tables, authored or reviewed drafts of the article, and approved the final draft.

Andrei G. Felice conceived and designed the experiments, performed the experiments, analyzed the data, prepared figures and/or tables, authored or reviewed drafts of the article, and approved the final draft.

Amanda A. Seribelli conceived and designed the experiments, performed the experiments, analyzed the data, authored or reviewed drafts of the article, and approved the final draft.

Dália P. Rodrigues conceived and designed the experiments, performed the experiments, analyzed the data, authored or reviewed drafts of the article, and approved the final draft.

Siomar C. Soares conceived and designed the experiments, performed the experiments, analyzed the data, authored or reviewed drafts of the article, and approved the final draft.

Marc W. Allard conceived and designed the experiments, authored or reviewed drafts of the article, and approved the final draft.

Juliana P. Falcão conceived and designed the experiments, authored or reviewed drafts of the article, and approved the final draft.

The following information was supplied regarding data availability:

The 98 genomes analyzed in the study are available at GenBank (Table 1).

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
