# Peer review of "Comparative genomics reveals high genetic similarity among strains of Salmonella enterica serovar Infantis isolated from multiple sources in Brazil"

_PeerJ, doi:10.7717/peerj.17306_

## Round 0.1 · original submission · Minor Revisions

I agree with most of the comments raised by the external reviewers. In particular, please be more explicit about the scale of diversity observed in your study, for example using % similarity. Also, please address review concerns about the over-representation of clinical isolates in specific clusters (ie GG1), and how this affects your conclusions. Ensure your manuscript adheres to accepted scientific nomenclature and abbreviations for Salmonella. The reviewers also make a several helpful suggestions for improving the clarity of your manuscript that are worth considering.

·

Basic reporting

Basic reporting: the scientific English language used by the authors is relatively good though there are a number of awkward sentences. The introduction and background are good with the exception that the authors fail to mention the increasing importance of serovar Infantis in the spread of multi-drug resistance, in particular the recently emerged clone (ESI) that was first identified in Israel (Aviv, et al, 2014, doi: 10.1111/1462-2920.12351) and has since spread around the world. The manuscript is structured well, all figures and tables are relevant and adequately labelled. All sequence data has been made publicly available.

Experimental design

Experimental design: The research is original and within the scope of the PeerJ. The research question the authors formulated is filling a knowledge gap about ser. Infantis in Brazil, a major meat exporter globally. However, I feel the authors could strengthen their dataset and the importance of this manuscript by building on their previously published work for these same 80 strains by adding the predicted AMR data from Vilela et al 2022a (DOI: 10.1111/jam.15430). Ser. Infantis is currently one of the most important Salmonella serovars spreading MDR around the world. The authors’ 2022a paper overlays the predicted AMR data with the PFGE data, but it would be hugely important to overlay the predicted AMR data over the wgMLST data, in particular the 5 different lineages seen in the wgMLST analysis. I would also suggest adding a representative of the recently emerged MDR ESI clone in the set of non-Brazilian sequences. Srednik et al 2023 (DOI 10.3389/fmicb.2023.1166908) list good candidate sequences.
Another thing that is missing from the experimental design is that the authors should use wgMLST better to describe diversity. It is hard to figure out the true amount of diversity from the tee alone in Fig 1. Please see comments related to this in Detailed comments no. 11
Methods are described with sufficient detail, for the most part.

Validity of the findings

Validity of findings: I do question to some extent the conclusion the authors make about the cross-contamination of the clinical and non-clinical sources. To me the distribution of the clinical (human) strains is not random among the 4 detected clusters/sub-lineages (see comment no. 25 in the Detailed comments). Also, to better asses diversity, the authors should annotate the wgMLST tree with allele ranges within the 5 clusters and allele differences between the clusters. There is probably more diversity in there based on wgMLST than what the other methods used (GeGenees, orthologous proteins, etc) imply. The conclusions related to the comparison to non-Brazilian strains would be stronger if the predicted AMR data was added to the investigation. While the Brazilian strains appear to cluster mainly separately from the non-Brazilian strains, I would say that the 18 non-Brazilian strains included in the study is a rather limited set and does not probably give us the entire picture on whether the Brazilian genotypes have spread outside Brazil. This should be mentioned as a limitation. It is curious though that 2 strains from the US and Mexico cluster right in the middle of wg cluster 1. This is where the allele differences would be very helpful: how similar are they really?

Additional comments

Detailed comments:

1. Line 28 and throughout the manuscript: “S. Infantis” is not approved Salmonella nomenclature. The genus Salmonella can only be shortened as “S.” if the species name enterica is also provided. Correct nomenclature is “S. enterica ser. Infantis” or “Salmonella Infantis” or simply “ser. Infantis”.
2. Lines 80-83: while AMR, and the importance of genomic epidemiology to monitor its spread is mentioned in this sentence, the rest of the study completely ignores this important and emerging feature of ser. Infantis. I highly suggest incorporating AMR prediction into this study.
3. Line 85: add “analysis” after “gene-by-gene”
4. Line 108: include the library prep method used and the minimum target coverage
5. Line 115: delete “Isolation”. The name of the NCBI pipeline is simply “Pathogen Detection”
6. Line 118: please include the origin (source country and isolation year) for the closed reference genome
7. Lines 132, 138, 142, 145, 151, 173, 181, 182, 195, 201 : “format” instead of “extension”
8. Lines 136, 146, 175, 186, 196 and 203: “The specifics” instead of “The particularities of the functioning”
9. Line 137: “have been” instead of “has been”
10. Line 199: add “of” after “genomes”
11. Lines 207-223 and Fig 1: It is hard to tell from the wgMLST tree how much diversity there is truly because there is no diversity scale (no. of allele differences or % of similarity) on the tree. Ideally, the tree should be annotated with allele ranges and differences, i.e., the allele range within each 5 clusters and the allele differences between each of the 5 clusters. Also summarize these differences with a couple of sentences over here in the result section.
12. Lines 210-218: instead of listing the exact numbers of isolates from each source for each cluster which makes this section very list-like and awkward maybe just mention that isolates from different sources were scattered among the 4 Brazilian clusters but then also mention if some sources were under- or over-represented. For example, it seems like clinical isolates were under-represented in clusters 3 and 4.
13. Lines 220-223: these sentences are awkward. Maybe state along the lines that the reference strain and the US strain 2013K-0515 did not fall into any cluster but were most closely genetically related to clusters 1, 2 and 3.
14. Lines 225 and 227: Fig 2 instead of Fig 1
15. Lines 227-234: the same comment as above for lines 210-218.
16. Lines 263-265: summarize the results from the figure with a sentence or two.
17. Line 301: delete “the” in front of “most”
18. Lines 321-323: this is an awkward sentence: ...that a uniform horizontal positioning of the LCBs was observed for…
19. Lines 329-330: “to contribute to bacterial virulence” instead of “to influence in bacterial virulence”
20. Lines 338-342: this is an awkward sentence: “However, limited information is available for NTS serovars about the role of several other phages detected in this study such as…”
21. Line 342: “contributing to the virulence” instead of “acting in the virulence”
22. Line 351: “were detected” instead of “have been detected”
23. Line 364: the reference strain Sinfa was isolated in 1973
24. Line 367: typo in “acquire”
25. Lines 390-396: while the overall similarity of the 80 Brazilian strains was high at least by Gegenees (cannot assess wgMLST without allele ranges and differences), to me it looks like the clinical (human) isolates are not randomly distributed among the 4 wgMLST and Gegenees clusters. The clinical isolates appear to be under-represented in wgMLST clusters 3 and 4 and in the Gegenees tree, they appear to be concentrated in the cluster GG1. Maybe further categorizing the types of foods and animals in these clusters would bring more clarity. To me it looks like there are sub-lineages that are not frequent in humans. And then it would be important to overlay the antimicrobial resistance data to these sub-lineages too and annotate Fig 1 with allele ranges/differences. The Infantis population in Brazil may not be as homogeneous as the authors imply.

·

Basic reporting

No comment

Experimental design

No comment

Validity of the findings

No comment

Additional comments

Lines 74 to 84 – I think the authors can rewrite the sentences in this paragraph to better convey the idea. If I understood correctly, the authors are saying that PFGE and MLST were considered gold standard techniques with their broad utility. However, now WGS is much more discriminative, fast, and cost-effective, leading to its replacement of these old techniques as an essential tool in the global tracking. Here is my suggestion to clarify the idea:

In the past, methods such as PFGE and MLST were considered gold standard techniques for studying NTS serovars, including S. Infantis strains in Brazil (Almeida et al., 2013; Monte et al., 2019; Vilela et al., 2022a). However, with the development of novel methods, broader access, and cost reductions in WGS, there have been significant advances in monitoring genomic relationships and antimicrobial resistance development among bacterial pathogens (Gilmour et al., 2013; Allard et al., 2018). As a result, WGS is now replacing these older techniques as an essential tool in the global tracking of zoonotic and foodborne pathogens of public health importance, thanks to its discriminative nature, speed, and cost-effectiveness (Gilmour et al., 2013; Allard et al., 2018). This shift towards WGS has been driven by international efforts to integrate human, animal, and environmental health through the One Health philosophy (Gilmour et al., 2013; Allard et al., 2018).

Lines 95 to 97 – It is true that S. Infantis is still very few explored in Brazil. However, there is a specific paper that I believe also provided an important contribution to this serovar in Brazil, and I encourage the authors to add it along with these important already cited papers. Please check if the publication (10.1016/j.meegid.2021.104934) also refers to the sentence: "However, in Brazil, genomic information is still scarce for this serovar (Monte et al., 2019; Melo et al., 2021; Bertani et al., 2022; Vilela et al., 2022ab)."

Line 98 to 102 – Consider improving the impact of your sentence. Suggestion:
“Given the limited genomic information available about this serovar in Brazil, the objectives of this study were to genetically characterize S. Infantis strains isolated from various sources such as food, farm and industry environments, humans, animals, and animal feed from 2013 to 2018 using comparative genomic analyses. Additionally, we aimed to establish correlations between these strains and isolates from other countries”.
Line 118 to 121 – I understand that the selection criteria were based on countries with the highest number of genomes deposited. However, it is still not clear how the specific criteria were applied within each country. For instance, while there are thousands of genomes from the USA and the UK, the authors only selected 4 genomes from the UK and 6 from the USA. Additionally, it is not clear based on which criteria the authors selected 25 strains for plasticity visualization and 5 strains for gene synteny analysis. Further clarification on the selection process would be beneficial to better understand the rationale behind the chosen genomes.

Line 164 – I believe accessory genome seems more appropriate than shared genome.

Line 178 and others – It is unnecessary to say “in the fna extension” or any other file extension.

Lines 275 -276 – (item 5.6 ???)

Lines 291 to 293 – This sentence has already been written in the introduction section. I suggest not repeating it in the discussion.

Lines 321 to 328 – Don't you think that adding genomes from countries other than Brazil in these analyses may enhance the robustness of demonstrating the conserved genomic structure among Brazilian strains? I believe that comparing draft genomes with a complete reference genome may introduce bias. I suggest including other draft genomes from countries other than Brazil.

Lines 329 to 336 – Yes, this is the reason why it may be interesting to add other draft genomes from other countries in these analyses.

Lines 371 to 377 – This affirmation corroborates with the necessity of comparing the gene synteny of Brazilian strains with more genomes from other countries instead of comparing with a single reference strain from 1963.

Lines 400 to 406 – Using various genomic approaches, the results presented here emphasize the significant genomic similarity among the studied strains, indicating that Salmonella Infantis is widely spread among both clinical and non-clinical sources in Brazil. Additionally, the data generated contribute to novel information regarding the genomic diversity of S. Infantis strains isolated in Brazil and provide a comparison with strains from other countries, indicating the occurrence of a potentially genetically distinct profile in Brazil.

---

## Round 0.2 · Major Revisions

Thank you for the submission of your revised manuscript. This version is significantly improved as a result of your work. Most of the suggestions pertain to minor changes in the document; however, I agree with the comments from Reviewer 1 regarding your conclusion that the current genomes represent a distinct Brazilian S. enterica serovar profile. Your choice of comparative genomes should include non-Brazilian sources within the same SNP cluster as your genomes, or you should explicitly state if no such genomes exist, in order to support this conclusion. Please also provide the NCBI ID or IDs for these SNP clusters; at last count there appear to be over 400 unique clusters of S. enterica from Brazil in the Pathogen Detection database, making it prohibitively difficult to conduct an independent assessment.

·

Basic reporting

No further comments

Experimental design

The resubmitted manuscript titled “Comparative genomics reveals high genetic similarity among strains of Salmonella enterica serovar Infantis isolated from multiple sources in Brazil” has improved significantly from its original version. My remaining main concerns relate to the experimental design and conclusions. The authors had added information about how the non-Brazilian genomes were selected to the study and if I understood it correctly, the selection process introduced significant bias to the study and hence to the conclusions (please see the specific comments #6 and #13 below).

Validity of the findings

Please see experimental design above and specific comments, particularly #12 and #13, below

Additional comments

1. Line 35: “assess” instead of “access”.
2. Lines 91-92: PFGE and MLST need to be spelled out since they are mentioned here for the first time.
3. Lines 136-138: The library prep method and the minimum target coverage need to be stated.
4. Line 138: add “annotated using” in front of “NCBI’s”…”.
5. Line 145: “were” duplicated.
6. Lines 147-148: isn’t the selection of the non-Brazilian genomes rather biased if only genomes from SNP clusters other than those containing the Brazilian strains were selected? That in essence means that closely related genomes were omitted on purpose which will affect the conclusions. In order for the authors to state/conclude that the Brazilian genotype is unique to Brazil, at least some of the non-Brazilian genomes should have been selected from the same SNP clusters (if there were any in the same clusters as the Brazilian strains. If there were not, then that needs to be stated in the methods that the Brazilian genomes formed their own SNP clusters on NCBI). NCBI’s SNP clusters are defined by 50 SNPs so even strains within the same cluster are not always that closely related, strains from other SNP clusters are light years away.
7. Lines 149-151: this is an awkward sentence, please rephrase: “the number of genomes per country was determined in proportion to the number of sequence uploads from a given country”.
8. Line 301: “similarity” duplicated.
9. Lines 305-310: instead of listing the detected resistance genes, please list the predicted antibiotic resistances for each group.
10. Line 345: delete “and”
11. Lines 409-411: this is an unclear sentence, please rephrase: “Regarding the distribution of the Brazilian genomes of Salmonella Infantis observed in wgMLST and Gegenees phylogenies, no clear correlation was observed between any specific lineage and the isolation source or date.”
12. Lines 411-414: this statement is not correct. All the multi-drug resistant strains appear to be in lineage 2. So even though there appears to be no correlation between lineage and isolation source, the MDR strains clearly have a different genetic background than the pan-susceptible/less resistant strains and that should be stated in the discussion.
13. Lines 433-435 and 544-546: I am not so sure you can state this if you on purpose hand-picked the non-Brazilian strains from NCBI so that they were from different SNP clusters than the Brazilian strains. If you want to see whether the Brazilian genotypes are present outside Brazil or not you should also pick non-Brazilian sequences from NCBI that appear to be closest to the Brazilian sequences, i.e., from the same NCBI SNP cluster.
14. Line 468: Salmonella nomenclature still not fixed
15. Line 506: “an even distribution” instead of “a regular distribution”
16. Line 508: A typo in “be”
17. Figure 1: I like the version of this figure in the supplemental figure 1 much better than this one included in the main paper. Why wouldn’t you use that one instead, just add the geographic locations to the strain IDs.
18. Lines 780-784, Figure 2: the matrix in the figure is unreadable because it is so small and when you zoom into it the numbers are all blurred. I assume the matrix displays the similarity percentages? If so, it would be helpful to add explanation in the figure legend for the colors seen in the matrix, i.e., green highly related, yellow less closely related and state the range for the percentages for both colors. I think it is important to clearly display the level of relatedness in this figure since Fig 1 does not really quantify the level of relatedness.
19. Line 787: add “contributing to the” in front of “development…”.

·

Basic reporting

No comment

Experimental design

No comment

Validity of the findings

No comment

Additional comments

No comment

---

## Round 0.3 · Major Revisions

Thank you for the submission of your revised manuscript. This version is significantly improved as a result of your work. Most of the suggestions pertain to minor changes in the document but ALL must be addressed.

Please pay particular attention, and answer adequately, to the recurrent remark of the review about the significance of the non-Brazilian genomic matches to the Brazilian genomes, and correct the sentences referred to in Reviewer comments 11 and 12.

Best regards

·

Basic reporting

Some sentences are awkward. In my detailed comments (under Additional comments), I have suggested how to improve them

Experimental design

No additional comments since my previous concerns have been adequately addressed.

Validity of the findings

The authors largely dismiss the significance of the non-Brazilian genomic matches to the Brazilian genomes. The fact that multiple genomes from North and South America and UK fall in the same wgMLST cluster with the Brazilian genomes suggests that some exportation of the Brazilian genotype has already occurred. This is a finding that needs to be emphasized, particularly because of the Brazilian strains carry some antimicrobial resistance genes, including some that are MDR.

Additional comments

The most recent modifications to the manuscript titled “Comparative genomics reveals high genetic similarity among strains of Salmonella enterica serovar Infantis isolated from multiple sources in Brazil” had again improved the manuscript substantially, particularly in bringing in clarity to how the non-Brazilian genomes were selected. I have a few more suggestions for edits to make the English language more fluent but also to emphasize in the discussion and conclusions the significance of the few genomic matches found to the Brazilian genomes outside Brazil.
1. Line 120: “established” instead of “stablished”
2. Lines 147-156: this is all said in a way too complicated manner. Don’t you just want to state that of the 18 non-Brazilian genomes included for comparison purposes, 8 shared SNP clusters with the 80 Brazilian study genomes, while another 8 came from SNP clusters with no Brazilian genomes in them. The remaining 2 were picked because they carried the epidemic pESI plasmid and did not share a SNP cluster with the 80 study genomes either.
3. Line 269: “the strain isolation sources” instead of “strains’ the isolation sources”
4. Line 293: weren’t there 5 subsets analyzed instead of 3?
5. Line 314: observe few what?
6. Line 340: “analyzed here” instead of “here analyzed”
7. Lines 348-349: “the circulation of this specific genomic profile mostly among animals in the country” instead of “the occurrence of this specific genomic profile circulating mostly among animals in the country”
8. Line 351: “revealed” instead of “evidenced”
9. Line 352-354: This sentence is very awkward. Please rephrase: “While profiles 1 and 3 grouped strains predicted to be less resistant, profile 2 contained strains with genetic markers predicting multi-drug resistance.”
10. Lines 355-358: this sentence is very awkward. Don’t you just want to state that while the wgMLST produced clustering between the Brazilian and non-Brazilian genomes that was similar to what was seen with the NCBI Pathogen Detection SNP analysis, the Gegenees clustering lacked the same resolution.
11. Lines 364-366: this statement is not true: the Gegenees clustering did not completely agree with the SNP or wgMLST clustering at the genome/strain level. It is clear that it has poorer resolution. You state that in the results lines 269-272.
12. Lines 370-372: this statement is not entirely true either: there were a few genomes from North and South America and UK that mixed in with the Brazilian genomes suggesting that, while the genotype might be of Brazilian origin, there has already been exportation of the genotype into other countries.
13. Line 381: “that an equal linear organization of LCBs was not observed” instead “that it was not observed an equal linear organization of the LCBs”
14. Lines 440-450: Another limitation is that the platform employed for the wgMLST analysis did not allow the annotation of the resulting tree with the actual allele differences or generation of an allele matrix, making it extremely difficult to evaluate how closely related the strains within each cluster truly were or how genetically distinct/tight the observed clusters were. Comparison to the Gegenees clustering is not a good measure because it is obvious from this data that the Gegenees as a method lacks discriminatory power.
15. Lines 474-478: this last sentence ignores the fact that non -Brazilian matches were found by both the wgMLST and SNP analyses which suggests that at least some exportation of the Brazilian genotype has already occurred.

---

## Round 0.4 · Minor Revisions

Dear Authors,
Your manuscript has been significantly improved and only a couple of minor alterations are requested.

Please address all these remaining points, in the revised version of the manuscript
Thank you very much!

·

Basic reporting

I have no further major concerns regarding the manuscript “Comparative genomics reveals high genetic similarity among strains of Salmonella enterica serovar Infantis isolated from multiple sources in Brazil”. I only have some minor edits to basic reporting.
1. Lines 54-58: this sentence is too long. Please split into two sentences.
2. Line 149: “than the ones” instead of “then the”
3. Line 198: “based on” instead of “based in”
4. Line 218: “included” instead of “comprised”
5. Line 251: “The remaining” instead of “All”
6. Line 312: delete “the”
7. Line 327: please clarify here that the SNP clusters you refer to are from the NCBI Pathogen Detection Pipeline
8. Line 350: “detected by wgMLST” instead of “differently than wgMLST”
9. Line 437: “inability” instead of “impossibility”
10. Lines 438-439: this sentence is awkward: please rephrase: “…complicated the inference of the genetic proximity among the strains within each cluster and between clusters.”
11. Lines 440-458: these two paragraphs do not belong over here under limitations. They belong to the sections discussing the phage and the pangenome analysis results, respectively.
12. Lines 462-463: “suggesting wide distribution of closely related genotypes among diverse sources in Brazil” instead of “suggesting its wide distribution among diverse sources in Brazil”
13. Line 467: “genotypes” instead of “genotype”

Experimental design

No further comments

Validity of the findings

No further comments

Additional comments

No further comments

---

## Round 0.5 · accepted · Accept

Thank you very much for the final minor revisions.
Your manuscript is now acceptable for publication.
Congratulations!